

# CST-Net: community-guided structural-temporal convolutional networks for popularity prediction

Xuxu Zheng[1,2], Peng Bao[3], Lin Qi[3], Chen Tian[3] and Huawei Shen[1,2]

[1] University of Chinese Academy of Sciences, Beijing, China
[2] Institute of Computing Technology, Chinese Academy of Sciences, Beijing, China
[3] Beijing Jiaotong University, Beijing, China

## ABSTRACT

The ability to predict the popularity of online contents has important implications in a wide range of areas. The challenge of this problem comes from the inequality of the popularity of content and the numerous complex factors. Existing works fall into three main paradigms: feature-driven approaches, generative models, and methods based on deep learning, each with known strengths and limitations. In this article, we propose an end-to-end deep learning framework, called CST-Net, to combat the defects of existing methods. We first learn a low-dimensional embedding for each user based on historic interactions. Then, users are clustered into communities based on the learned user embeddings, and information cascades are represented as a series of episodes in the form of community interaction matrix. Afterwards, a convolutional architecture is applied to learn the representation of the entire information cascade. Finally, the extracted structural and temporal features are further combined to predict the incremental popularity. We validate the effectiveness of the proposed CST-Net by applying it on two different types of population-scale datasets, *i.e.*, a microblogging dataset and an academic citation dataset. Experimental results demonstrate that the proposed CST-Net model consistently outperforms the existing competitive popularity prediction methods.

## INTRODUCTION

The emergence of Web 2.0 applications brings the explosive growth of user generated contents (UGC). It is significantly important to predict the future popularity of UGC items, such as microblogs, academic articles, and videos. Popularity prediction has important implications in many domains, including viral marketing (*Leskovec, Adamic & Huberman, 2007*), public opinion monitoring (*Watts & Dodds, 2007*), *etc*. However, popularity prediction is challenging since numerous factors can affect the asymmetric and broadly-distributed popularity of online content.

Recently, great efforts have been made to study the popularity prediction on social networks. In general, current models fall into three main paradigms, each with known strengths and limitations. Some methods focus on exploring relevant features and

Corresponding author
Peng Bao, baopeng@bjtu.edu.cn

generally making predictions in a supervised framework with machine learning algorithms (*Szabo & Huberman, 2010*; *Bao et al., 2013*; *Shulman, Sharma & Cosley, 2016*; *Hong, Dan & Davison, 2011*). These feature-driven approaches devote to verifying the effectiveness of a bag of pre-defined interpretable features. However, they heavily rely on hand-craft features based on prior knowledge. Meanwhile, there are still numerous factors to be investigated and it is not flexible to apply them to a broad spectrum of data domains. Afterwards, some researchers treat the popularity dynamics as time series, making predictions by modeling the process through which individual items gain their attentions (*Rodriguez, Leskovec & Schölkopf, 2013*; *Shen et al., 2014*; *Zhao et al., 2015*; *Bao et al., 2015*). Despite their initial success in leveraging fine-grained timing information in the event series, a major limitation of these existing studies is that they often draw various parametric assumptions about the latent dynamics governing the generation of the observed temporal point process. In recent years, there has been heightened research interest regarding the end-to-end deep learning framework for popularity prediction based on deep learning (*Bourigault et al., 2014*; *Du et al., 2016*; *Li et al., 2017*; *Cao et al., 2017*; *Xiao et al., 2017*; *Dou et al., 2018*). This line of works can automatically learn the valuable information from raw data. However, existing works mainly focus on sampling sequences from cascade graphs and then feeding these sequences into neural networks, which have paied heavy attention on the microscopic level of information cascades. The way of learning the representation of the entire information cascade directly still remains unresolved.

Inspired by *Zhang, Zheng & Qi (2017)* which focus on employing a deep residual neural network framework for citywide crowd flows prediction, we aim to model the information diffusion from the mesoscopic perspective of communities. More specifically, information diffusion is treated as the inflow and outflow among different communities. According to the empirical findings in *Bao et al. (2013)*, the structural characteristics of diffusion cascade at an earlier time can help predict its final popularity. For example, if a tweet has been spread to different communities of the network, it is more likely to become known by a greater population in the future.

In this article, we propose community-guided structural-temporal convolutional networks, called CST-Net, to combat the defects of existing methods, leveraging an end-to-end deep learning framework for popularity prediction. Firstly, we learn a low-dimensional embedding for users based on their historic interactions. With the learned user embeddings, users are clustered into communities, and information cascades are represented as a series of episodes in the form of community interaction matrix. Then, a convolutional architecture is applied to learn the representation of the entire information cascade. Finally, the extracted structural and temporal features are further combined to predict the incremental popularity. We validate the effectiveness of CST-Net by applying it on two different types of population-scale datasets, *i.e.*, Sina Weibo and APS. Experimental results demonstrate that the proposed CST-Net model consistently outperforms the existing competitive popularity prediction methods.

The main contributions of this article are three-fold:

- We employ convolution-based residual networks to simultaneously model local and global structural dependencies among users in information diffusion processes from the mesoscopic perspective of communities.
- We represent information cascade as a series of episodes in the form of community interaction matrix, leveraging the temporal properties of recent time, near history and distant history during the dynamic process.
- We evaluate our proposed model on two different types of population-scale datasets. The experimental results demonstrate the advantages of our model compared with four baselines.

The rest of this article is organized as follows. The literature review for popularity prediction is detailed in "Related Works". "Preliminaries" introduces the definition of popularity prediction problem and background about the convolutional neural networks. "Model" describes the three components of our proposed model, followed by comprehensive experiments using two population-scale datasets to evaluate the performance of our proposed model in "Experiments". Conclusion and future directions are given in the end.

## RELATED WORKS

Existing approaches for popularity prediction can be classified into three categories: feature-driven approaches, generative models, and methods based on deep learning.

### Feature-driven approaches

Great efforts have been made to investigate an extensive set of hand-crafted features for predicting the popularity of online contents, including tweets (*Hong, Dan & Davison, 2011*; *Zhao et al., 2015*), microblogs (*Bao et al., 2013*; *Bian, Yang & Chua, 2014*), videos (*Pinto, Almeida & Goncalves, 2013*; *Chang et al., 2014*; *Ding et al., 2015*), academic articles (*Shen et al., 2014*; *Wang, Song & Barabási, 2012*), to name a few. Most of the studies devote to identifying temporal properties (*Szabo & Huberman, 2010*; *Pinto, Almeida & Goncalves, 2013*), structural features of the cascade at the early stage (*Bao et al., 2013*; *Cheng et al., 2014*), and the content features (*Hong, Dan & Davison, 2011*; *Ma, Sun & Cong, 2013*) as the most predictive factors. A classic approach is to making predictions by applying regression models (*Szabo & Huberman, 2010*; *Pinto, Almeida & Goncalves, 2013*; *Martin et al., 2016*) or classification models (*Shulman, Sharma & Cosley, 2016*; *Cheng et al., 2014*). *Szabo & Huberman (2010)* found that the final popularity is reflected by the popularity in early period by investigating Digg and YouTube. A direct extrapolation method is then employed to predict the long-term popularity. Later, *Cheng et al. (2014)* discovered that temporal features are most predictive of a cascade's final popularity. Besides, user features, particularly user's influence in past are shown to be informative predictors (*Bakshy et al., 2011*; *Feng et al., 2018*). Recently, *Martin et al. (2016)* explored the limits of predictability

in complex social systems, suggesting that even with unlimited data predictive performance would be bounded well below deterministic accuracy. *Li et al. (2020)* explored group-level features to predict the popularity of new Meetup groups. In general, these feature-driven approaches devote to verifying the effectiveness of a bag of hand-crafting features in order to learn a better prediction model. However, there are still numerous factors to be investigated and it is not flexible to apply them to a broad spectrum of data domains. Moreover, feature-driven approaches heavily rely on the quality of artificially designed features, which are hand-crafted based on human prior domain knowledge.

## Generative models

The other line of enquiry, in contrast, treats the popularity dynamics as time series, making predictions by modeling the process through which individual items gain their attentions (*Shen et al., 2014*; *Gao, Ma & Chen, 2015*; *Crane & Sornette, 2008*; *Lerman & Hogg, 2010*). *Crane & Sornette (2008)* studied the relaxation response of a social system after endogenous and exogenous bursts of activity using the time series of daily views on YouTube, finding that most activity can be described accurately as a Poisson process. Subsequently, *Shen et al. (2014)* employed reinforced Poisson process to model the arriving process of article citations. Recently, Hawkes self-exciting process was employed to captures the triggering effect of each attention (*Zhao et al., 2015*; *Bao et al., 2015*; *Mishra, Rizoiu & Xie, 2016*; *Bao & Zhang, 2017*). Despite their initial success in leveraging fine-grained timing information in the event series, however they make stronger assumptions about the diffusion process. *Mishra, Rizoiu & Xie (2016)* proposed a marked Hawkes self-exciting point process, which intuitively aligns with the social factors responsible for diffusion of cascades: social influence of users, social memory and inherent content quality. Furthermore, *Rizoiu et al. (2017, 2018)* investigated the correlation between the endogenous response and the exogenous stimuli of a social system, and explored the connection between Hawkes point processes and SIR epidemic models. *Xu et al. (2023)* enabled integration of structural and temporal information in a diffusion process. Generative approaches learn the inherent characteristics of online content and propagation mechanism on different communication platforms, which achieve good interpretability. However, most generative approaches generally depend on specific assumptions that are unknown in reality so that their representation ability are relatively limited.

## Deep learning methods

In order to deal with such complex popularity dynamics and automatically leverage predictive information from raw data, in recent year, there has been heightened research interest regarding the popularity prediction methods based on deep learning (*Bourigault et al., 2014*; *Du et al., 2016*; *Li et al., 2017*; *Cao et al., 2017*; *Xiao et al., 2017*; *Zhao, Zhang & Feng, 2022*; *Bao, Yan & Yang, 2024*). *Du et al. (2016)* viewed the intensity function of a temporal point process as a nonlinear function of the history, and employed a recurrent neural network (RNN) to automatically learn a representation of influences from the event history. Later, *Xiao et al. (2017)* modeled both background and history effect by two RNN,

respectively. The recent successes of deep learning in a wide range of areas also inspires some end-to-end deep learning framework for popularity prediction. *Li et al. (2017)* presented DeepCas model that learn the representation of cascade graphs in an end-to-end manner, which significantly improve the performance of popularity prediction over strong baselines. Subsequently, *Cao et al. (2017)* proposed DeepHawkes model, which inherits the high interpretability of Hawkes process and possesses the high predictive power of deep learning methods, bridging the gap between prediction and understanding of information cascades. This line of works can automatically learn the valuable information from raw data. More recently, *Zhao, Zhang & Feng (2022)* utilized the temporal and structural information of cascade networks as input to predict the future growth of information cascades. *Bao, Yan & Yang (2024)* proposed a learning framework for popularity prediction *via* modeling both the temporal evolution in a separate snapshot and the inherent temporal dependencies among different snapshots based on the dynamic evolution process. However, existing works mainly focus on sampling sequences from cascade graphs and then feeding the sequences into recurrent neural networks. In this article, we take advantage of both the effective information demonstrated by traditional methods and the power of end-to-end deep learning framework for popularity prediction. Existing methods mostly utilize the order relationship between forwarding users to model the sequential information, while ignoring the impact of temporal features that mainly dominates the future popularity.

## PRELIMINARIES

In this section, we first formally formulate the popularity prediction problem, and then briefly introduce the convolutional neural networks before presenting the proposed model.

### Problem definition

Let $\mathcal{M}$ denote a set of items, *e.g.*, microblogs or academic articles. Suppose the set $\mathcal{M}$ contains $M$ items, it is denoted by $\mathcal{M} = \{m^i\}(1 \leq i \leq M)$.

**Definition 1 (cascade)** *For each item $m^i$, a cascade is characterized by $\mathcal{C}^i = \{(u_k^i, v_k^i, t_k^i)\}$, where the tuple $(u_k^i, v_k^i, t_k^i)$ corresponds to the k-th forwarding, meaning that user $v_k^i$ forwards the item $m^i$ from user $u_k^i$ at time $t_k^i$.*

**Definition 2 (popularity prediction)** *Given the cascades in the observation time window $[0, T)$, we aim to predict the incremental popularity $\Delta N_T^i$ between observed popularity $N_T^i$ and final popularity $N_\infty^i$ of each cascade $\mathcal{C}^i$, where $\Delta N_T^i$ is denoted as $\Delta N_T^i = N_\infty^i - N_T^i$.*

In the definition, $T$ indicates the earliness of the prediction and refers to the length of training period. It is worth noting that we predict the incremental popularity instead of the final popularity to avoid the intrinsic correlation between the observed popularity and the final popularity, which forms a more challenging scenario for popularity prediction.

### Convolutional neural networks

Convolutional neural networks (CNNs) have proven to be effective models for tackling a variety of visual tasks. For each convolutional layer, a set of filters are learned to express

local spatial connectivity patterns along input channels. By stacking a series of convolutional layers interleaved with non-linearities and downsampling, CNNs are capable of capturing hierarchical patterns with global receptive fields as powerful image descriptions. Recently, residual learning allows such networks to have a deeper structure (*He et al., 2016a*), which gains state-of-the-art results on multiple challenging recognition tasks.

Formally, a residual unit with an identity mapping is defined as:

$$\mathbf{X}_{l+1} = \mathbf{X}_l + \mathscr{F}(\mathbf{X}_l, \mathscr{W}_l), \tag{1}$$

where in the Eq. (1) $\mathbf{X}_l$ and $\mathbf{X}_{l+1}$ are the input and output of the $l$-th residual unit respectively; $\mathscr{F}$ is a residual function, *e.g.*, a stack of two $3 * 3$ convolution layers; $\mathscr{W}_l$ is a set of weights (and biases) associated with the $l$-th residual unit (*He et al., 2016b*).

To improve the ability of representation of a network by explicitly modeling the interdependencies between the channels of its convolutional features, *Hu, Shen & Sun (2018)* introduce a new architectural unit, which is termed as the "Squeeze-and-Excitation" (SE) block.

## MODEL

This section introduces the proposed deep Community-guided Structural-Temporal convolutional Networks for popularity prediction, called CST-Net. Figure 1 presents the main architecture of the proposed CST-Net, which takes the observed diffusion episodes of each cascade as input and outputs the final incremental popularity. Note that a diffusion episode is a set of users who adopt action in chronological order and it is a subset of the entire cascade graph. Therefore, it corresponds to a sequence of users who forwards a specific microblog or cites an academic article.

The main part of the proposed CST-Net model consists of three components: (1) user embedding, learning a low-dimensional embedding for each user based on historical interactions; (2) community interaction matrix construction, clustering users into communities based on the learned representations and constructing the community interaction matrix; and (3) convolutional architecture, feeding the community interaction matrix into convolutional neural networks to learn the representation of the observed cascade graph. Finally, we combine the learned representation of cascade graph with the extracted structural and temporal features to predict the incremental popularity. Detailed descriptions are given in the following sections.

### User embedding

Feature-based approaches have demonstrated that user features, such as, number of fans and past success, are effective for popularity prediction (*Shulman, Sharma & Cosley, 2016*; *Cheng et al., 2014*). However, it is still unclear which features is the best important for us to represent and distinguish users or users' influence. Recently, with the development of node embedding technology, learning users' low-dimensional representations directly from history data has been widely adopted. Therefore, in this article, we propose to learn users'

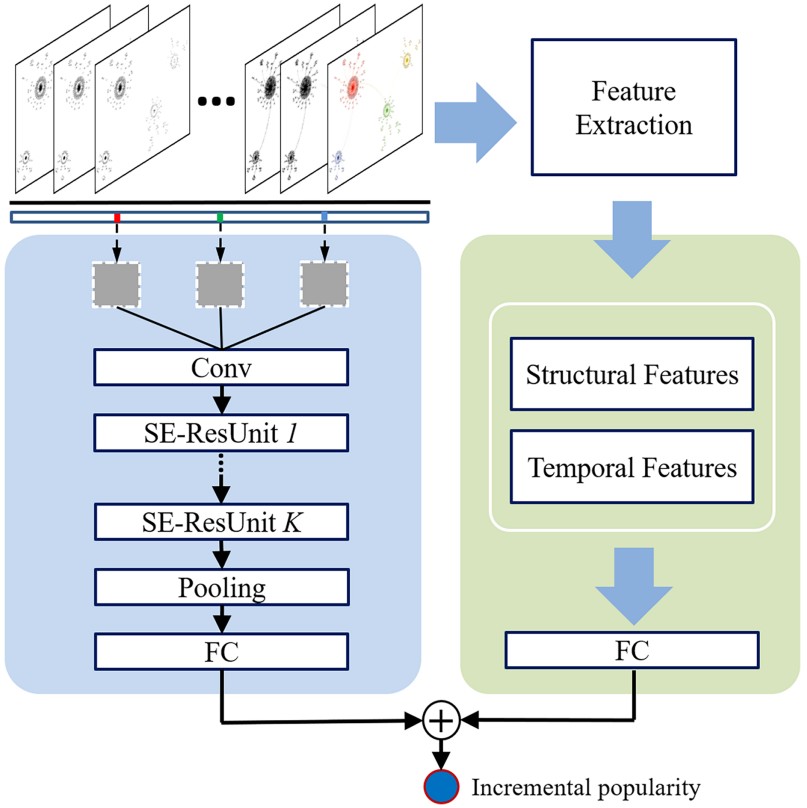

**Figure 1 Architectural overview of the proposed CST-Net.**

representations by using their historic interactions, which are regarded as the input feature vectors for the following community detection algorithm.

Generally, the static social graph among users are treated as input to learn their representations. However, it is usually intractable to obtain the entire social graph structure. Moreover, previous works have proved that historic interactions among users are more effective for measuring the influence and susceptibility among them (*Aral & Walker, 2012*). Hence, instead of social graph, we construct the interaction graph $G = (V, E)$ directly from historic interactions, where $V$ is the set of users and $E$ is the set of edges. Note that each edge in $E$ represents a retweet in microblogging network or a citation in academic network respectively.

Specifically, we employ Node2Vec (*Grover & Leskovec, 2016*) to characterize each user, which is a framework with outstanding performance for the continuous feature representation of nodes in large scale networks. The learned user embeddings are regared as input vectors for constructing community interaction matrix.

## Community interaction matrix construction

In this article, we aim to model information diffusion from the mesoscopic perspective of communities. We suppose that information propagation between users corresponds to the

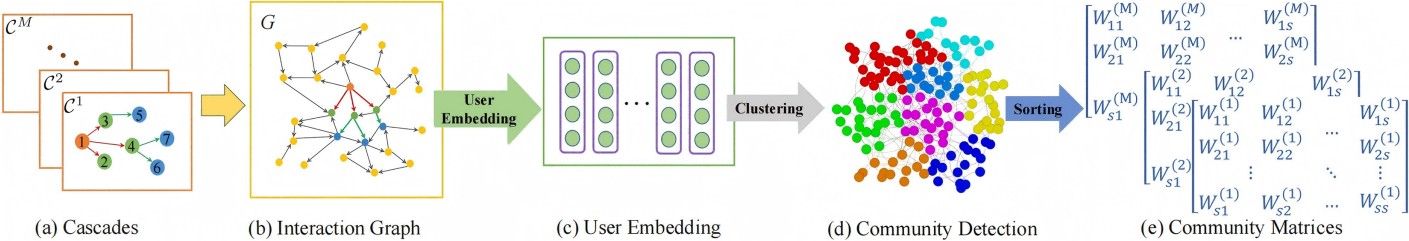

**Figure 2 Transformation of diffusion cascades to community interaction matrix.**

diffusion among different communities. As shown in Fig. 2, the observed episode of each cascade is transformed to the form of community interaction matrix. Specifically, information diffusion is regarded as the inflow and outflow among different communities. Therefore, it is significantly important for us to firstly detect the communities among users. In previous section, we obtain the learned user embeddings from user interaction graph. Here, we exploit the mini-batch k-means++ proposed by *Arthur & Vassilvitskii (2007)* to cluster users into $s$ communities. The community list is represented as $L$ and $s$ is the number of communities. The approach can handle large-scale networks with low computational complexity and has superior performance in community detection.

Given a cascade, it can be represented as the form of community interaction matrix $\mathbf{W} \in \mathbb{R}^{s*s}$, where the $(j, k)$-th value denotes the number of forwardings from community $j$ to community $k$ within the observation time window. Therefore, the problem of predicting the incremental popularity can be transformed into forecasting the information flow among different communities.

Now, we have obtained the $s$ detected communities among users based on their historic interactions. It is important and challenging for the downstream task to organize the community interaction matrix $\mathbf{W}$. When communities are arranged randomly in the matrix, the similarity between adjacent communities is uncertain and neglected obviously in the community interaction matrix $\mathbf{W}$. To extract deep features from user interactions, it is necessary to acquire the local proximity of the input community interaction matrix $\mathbf{W}$. Therefore, it is significantly important to assemble the community interaction matrix appropriately, which makes these communities with more interactions to get closer in the matrix $\mathbf{W}$. In this article, we propose an algorithm to assemble communities, preserving the proximity among them. Details are shown in Algorithm 1. Firstly, based on the obtained detected communities, the largest community is chosen and denoted by $L_m$, where $m = \lceil \frac{s}{2} \rceil$. According to some similarity measurement for communities, we obtain two other communities which are most similar with $L_m$, denoted by $L_{m-1}$ and $L_{m+1}$ respectively. Then, we continue to choose the most similar community in the rest and add it to the list denoted by $L_{m-2}$. Similarly, we add another one to the list denoted by $L_{m+2}$. The process continues until all communities are added to the list $L$. Finally, we reassemble the community interaction matrix $\mathbf{W}$ according to the order in the list $L$.

---

**Algorithm 1  Constructing community interaction matrix.**

**Input:** The detected community set $S$

**Output:** The community interaction matrix $\mathbf{W}$

1: Initialize an ordered community list $L$

2: Choose the largest community, add it to list $L$, denoted by $L_m$, where $m = \lceil \frac{s}{2} \rceil$, and remove it from $S$

3: Choose two communities most similar to $L_m$, add them to list $L$, denoted by $L_{m-1}$ and $L_{m+1}$, and remove them from $S$

4: **for** $k = 0$ to $m - 1$ **do**

5:      Choose the most similar community with $L_{m-k}$ and $L_{m-k-1}$

6:      Add it to list $L$, denoted by $L_{m-k-2}$ and remove it from $S$

7:      Choose the most similar community with $L_{m+k}$ and $L_{m+k+1}$

8:      Add it to list $L$, denoted by $L_{m+k+2}$ and remove it from $S$

9: **end for**

10: Reassemble the community interaction matrix $\mathbf{W}$ according to the order in the List $L$

11: **return** the community interaction matrix $\mathbf{W}$

---

For simplicity, in this article, the similarity between two communities $i$ and $j$ is defined as:

$$s(i, j) = U_{ij}/D_i^{out} + U_{ji}/D_j^{out} + U_{ij}/D_j^{in} + U_{ji}/D_i^{in},  \tag{2}$$

where in the Eq. (2) $U_{ij}$ represents the historic interaction from community $i$ to community $j$, such as, the number of retweets or citations from community $i$ to community $j$. $D_i^{out}$ and $D_i^{in}$ represent the outdegree and indegree of community $i$, respectively. Note that other similarity measurements can also be adopted. Here we treat investigating the effect of different similarity measurements as a future work.

## Convolutional architecture

Once we have obtained the community interaction matrix $\mathbf{W}$, an effective convolution neural network can be applied to extract the deep features of a specific cascade within the observation time window. In the proposed CST-Net, we utilize the convolutional neural network framework ResNet (*He et al., 2016a*) to predict the future incremental popularity. The "Squeeze-and-Excitation" block proposed by *Hu, Shen & Sun (2018)* is adopted to improve the performance of our model as a new architectural unit.

To capture the temporal features of a cascade, we divide the observation time window into three fragments, denoting recent time, near history and distant history. Then, they are fed into three channels of the convolution neural network. Moreover, we build multiple labels for each cascade which denotes the incremental popularity during the observation time window. Afterwards, we can obtain the learned representation for each cascade.

In addition, we extract some predictive structural and temporal features for each cascade. More specifically, we extract the structural features of the observed cascade as a measure of centrality and density, such as the number of leaf nodes, the number of first

layer nodes, average and max length of retweet or citation path. To capture the temporal dynamics of each cascade, we extract the mean time interval between each retweet or citation, the time latency of the first retweet or citation, the cumulative popularity, incremental popularity every 10 min for Sina Weibo and every 1 year for APS.

Finally, we feed the learned representation of each cascade graph and the aforementioned extracted features, denoted by $\mathbf{h^i}$, into a fully-connected layer (Eq. (3)) to predict the incremental popularity:

$$\Delta N_T^i = FC(\mathbf{h^i}). \tag{3}$$

## EXPERIMENTS

In this section, we present comprehensive experiments using two population-scale datasets to evaluate the prediction performance of the proposed CST-Net model. And an ablation study is also conducted to investigate the effectiveness of the components of CST-Net.

### Datasets

In order to demonstrate the effectiveness and generalizability of the proposed CST-Net, we evaluate it on two scenarios of popularity prediction. One of the scenario is forecasting the incremental popularity of retweet cascades in a social network and the other is predicting the incremental size of article citation cascades. In this article, the data were collected as previously described in DeepHawkes (*Cao et al., 2017*), where retweets and citations were considered as cascades, respectively.

**Sina Weibo dataset:** This data is collected from the most popular micro-blogging service in China, namely Sina Weibo, which has more than 500 million registered users and generates about 100 million messages per day. It consists of all the original microblogs that were submitted on June 1, 2016, and for each microblog, we collect its retweets within the next 24 h. In Sina Weibo, a cascade is generated by an original tweet and all of its retweets, and its popularity is the number of retweets. Figure 3A shows that the retweet cascade popularity exhibits a power-law distribution.

Generally, the life cycle of a microblog is relatively short. In Fig. 4A, we can see that the retweet number within 24 h $N_{24hours}$ after publication have a good approximation to the final popularity $N_\infty$ for each microblog. In our experiment, we set $T$, the observation time window of a cascade since the original microblog was released, from a range of 1, 2 and 3 h. For each $T$, we only consider the cascades with no less than 10 retweets and no more than 1,000 retweets in the observation window. Figure 5 shows that due to the diurnal rhythm of users, microblogs released in different period may have different popularity dynamic patterns. Therefore, we only consider microblogs published between 8:00 and 18:00. We split the remaining cascades into training set, validation set and test set by the first 70%, the middle 15% and the last 15% according to the publication time. Detailed statistics of the preprocessed dataset are shown in Table 1.

**APS dataset:** It comprises the articles published in all journals in American Physical Society from 1893 to 2009, consisting of 245,365 authors, 463,344 articles, and 4,692,026

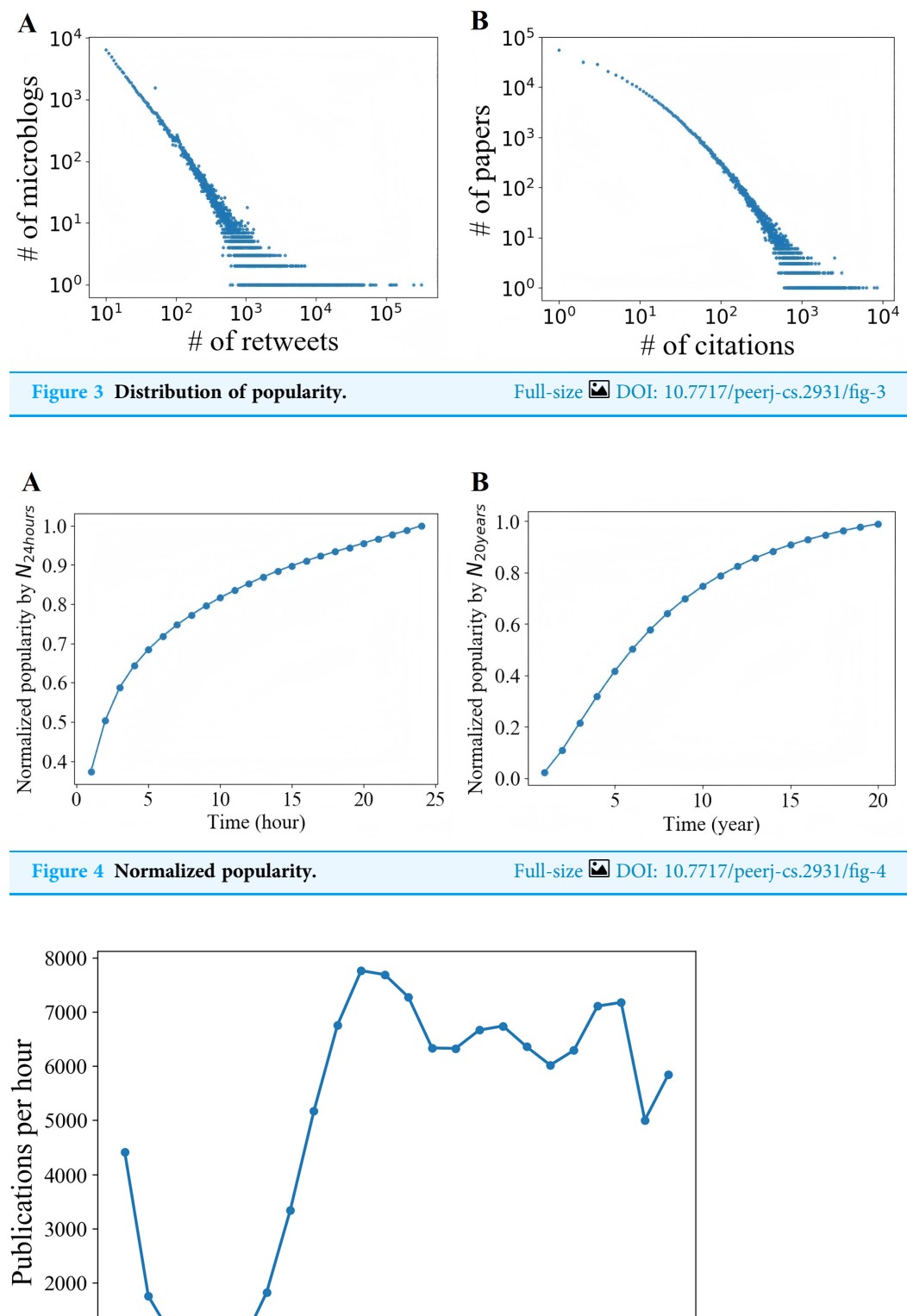

**Figure 3 Distribution of popularity.**

**Figure 4 Normalized popularity.**

**Figure 5 Diurnal rhythm of activities in Sina Weibo.**

**Table 1  Statistics of the datasets.**

| | | Sina Weibo | | | APS | | |
|---|---|---|---|---|---|---|---|
| **T** | | **1 h** | **2 h** | **3 h** | **3 y** | **6 y** | **9 y** |
| *Cascades* | Train | 28,597 | 33,886 | 36,496 | 15,160 | 30,497 | 35,795 |
| | Val | 6,110 | 7,254 | 7,828 | 3,281 | 6,736 | 8,020 |
| | Test | 6,125 | 7,283 | 7,840 | 3,288 | 6,751 | 7,983 |
| $Avg.N_T^i$ | Train | 62.92 | 69.15 | 71.95 | 23.36 | 32.15 | 39.55 |
| | Val | 66.83 | 73.54 | 76.63 | 24.22 | 37.54 | 48.63 |
| | Test | 61.84 | 67.65 | 71.94 | 26.66 | 40.61 | 52.20 |

**Note:**
   $Avg.N_T^i$ scales the actual popularity of the cascade size.

citations. For each article, the dataset includes title, DOI, PACS code, date of publication, names and affiliations of authors, a list of the articles cited, and so on. Similar with the preprocessing in DeepHawkes (*Cao et al., 2017*), we consider all coauthors of a article as an author group, which is analogy to a distinct person in the scenario of Sina Weibo. In APS, the cascade can represent the citation relationship of a article. In Fig. 3B, the citation cascade popularity also follows a power-law distribution.

Similar to Sina Weibo, as shown in Fig. 4B, for each article, we utilize the citation count within 20 years $N_{20years}$ to represent the final popularity $N_\infty$ for each article. Accordingly, we only use articles published between 1893 and 1989. In APS dataset, we choose the observation time window from a range of 3, 6 and 9 years. For each observation time $T$, only those articles with more than 10 citations is retained. Finally, on the basis of the publication time, we take the first 70% as training set, the middle 15% as validation set and the last 15% as test set. Table 1 shows the statistics of the processed dataset.

## Baselines

To evaluate the accuracy of our predictive model, we compare the proposed CST-Net model with state-of-the-art approaches. Specifically, the comparison methods in our experiments are listed as follows:

**Features:** Based on recent studies of popularity prediction, structural and temporal features provide strong evidence for the final popularity increment. Therefore, we extract all predictive features that could be generalized across datasets. These features include:

- *Structural features*. While it is often impossible to obtain the entire social network structure, we only extract the structural features of the observed cascade. We use the number of leaf nodes, the number of first layer nodes, average and max length of retweet or citation path as a measure of centrality and density.
- *Temporal features*: To capture the temporal dynamics of a cascade, We extract the mean time interval between each retweet or citation, the time latency of the first retweet or citation, the cumulative popularity, incremental popularity every 10 min for Sina Weibo and every 1 year for APS.

Once all the above predictive features are collected, we feed these feature vectors into a linear regression model with L2 regularization and a multi-layer perceptron (MLP), denoted as Feature-linear and Feature-deep, respectively.

**DeepCas** (*Li et al., 2017*) is state-of-the-art deep learning method for popularity prediction, which automatically learns the representation of individual cascade graph as a whole in an end-to-end manner and predicts the future incremental popularity.

**DeepHawkes** (*Cao et al., 2017*) leverages end-to-end deep learning to make an analogy to interpretable factors of Hawkes process, which inherits the high interpretability of Hawkes process and possesses the high predictive power of deep learning methods.

**CasCN** (*Chen et al., 2019*) conceptualizes the cascade graph as a discrete sequence of subgraphs, which introduces a graph convolutional network designed to capture local structural information within each snapshot, while recurrent neural networks are used to learn sequential information.

**GTGCN** (*Yang et al., 2022*) utilizes a temporal encoder to process timestamp information and applies an improved graph convolutional network to capture both structural and temporal information, combining temporal encoding with a gated recurrent unit to capture temporal dependencies across multiple snapshots.

## Evaluation metric

In order to validate the prediction performance of the CST-Net model, we compare it with the existing prediction models, in terms of two metrics: mean square log-transformed error (*MSLE*) and mean relative log-transformed error (*MRLE*), following the practice in previous work of popularity prediction (*Li et al., 2017*; *Cao et al., 2017*). Let $\Delta \hat{N}_T^i$ be the predicted incremental popularity for item $i$ up to time $T$, and $\Delta N_T^i$ be the observed incremental popularity. The *MSLE* (Eq. (4)) measures the average deviation between the predicted and observed incremental popularity over all items. For a dataset of $M$ items, the *MSLE* is defined as:

$$MSLE = \frac{1}{M} \sum_{i=1}^{M} \left( \log \Delta \hat{N}_T^i - \log \Delta N_T^i \right)^2. \tag{4}$$

The *MRLE* (Eq. (5)) expresses how large the absolute error is compared with the ground truth, which is defined as:

$$MRLE = \frac{1}{M} \sum_{i=1}^{M} \left[ \left( \log \Delta \hat{N}_T^i - \log \Delta N_T^i \right) / \log \Delta N_T^i \right]. \tag{5}$$

## Running environment and parameter settings

All experiments were done on a Ubuntu 20.04 server with Intel® Core™ i9-10850K CPU @ 2.60 GHz and a RAM of size 128 Gb. Besides, We use an NVIDIA 3080Ti GPU.

For hyper parameters, the mini-batch size of the stochastic gradient descent is set as 32 and 128 in Sina Weibo and APS dataset respectively. We select the initial learning rate from $\{0.1, 0.05, 0.001, \ldots, 10^{-5}\}$ and weight decay from $\{0, 0.01, 0.005, \ldots, 10^{-4}\}$. For

**Table 2 MSLE and MRLE of CST-Net and baseline methods.** Bold Entries represent the best performance among all the methods.

| | Sina Weibo | | | | | | APS | | | | | |
|---|---|---|---|---|---|---|---|---|---|---|---|---|
| T | 1 h | | 2 h | | 3 h | | 3 y | | 6 y | | 9 y | |
| Evaluation metric | MSLE | MRLE | MSLE | MRLE | MSLE | MRLE | MSLE | MRLE | MSLE | MRLE | MSLE | MRLE |
| Feature-linear | $3.633 \pm 1.1$ | $0.336 \pm 0.13$ | $2.834 \pm 0.9$ | $0.337 \pm 0.14$ | $2.305 \pm 0.8$ | $0.325 \pm 0.13$ | $3.776 \pm 1.2$ | $0.247 \pm 0.09$ | $3.136 \pm 1.0$ | $0.341 \pm 0.13$ | $2.745 \pm 0.8$ | $0.434 \pm 0.15$ |
| Feature-deep | $3.360 \pm 0.8$ | $0.338 \pm 0.12$ | $2.561 \pm 0.7$ | $0.347 \pm 0.12$ | $2.185 \pm 0.6$ | $0.356 \pm 0.14$ | $3.024 \pm 0.8$ | $0.223 \pm 0.08$ | $2.797 \pm 0.7$ | $0.328 \pm 0.12$ | $2.703 \pm 0.8$ | $0.433 \pm 0.14$ |
| DeepCas | $2.887 \pm 0.7$ | $0.364 \pm 0.14$ | $2.637 \pm 0.8$ | $0.394 \pm 0.13$ | $2.522 \pm 0.8$ | $0.417 \pm 0.15$ | $2.980 \pm 0.8$ | $0.286 \pm 1.02$ | $2.949 \pm 0.8$ | $0.435 \pm 0.15$ | $2.914 \pm 0.9$ | $0.546 \pm 0.16$ |
| DeepHawkes | $2.316 \pm 0.6$ | $0.293 \pm 0.09$ | $2.130 \pm 0.7$ | $0.312 \pm 0.11$ | $2.043 \pm 0.6$ | $0.342 \pm 0.12$ | $\mathbf{2.631 \pm 0.7}$ | $\mathbf{0.227 \pm 0.07}$ | $2.407 \pm 0.6$ | $0.326 \pm 0.12$ | $2.342 \pm 0.6$ | $0.425 \pm 0.13$ |
| CasCN | $2.273 \pm 0.6$ | $0.285 \pm 0.09$ | $2.122 \pm 0.9$ | $0.308 \pm 0.12$ | $2.112 \pm 0.7$ | $0.331 \pm 0.13$ | $2.893 \pm 0.8$ | $0.275 \pm 0.09$ | $2.627 \pm 0.7$ | $0.413 \pm 0.13$ | $2.831 \pm 0.7$ | $0.519 \pm 0.14$ |
| GTGCN | $2.265 \pm 0.7$ | $0.279 \pm 0.08$ | $2.117 \pm 0.7$ | $0.301 \pm 0.11$ | $2.011 \pm 0.6$ | $0.326 \pm 0.11$ | $2.811 \pm 0.7$ | $0.262 \pm 0.09$ | $2.594 \pm 0.7$ | $0.402 \pm 0.12$ | $2.721 \pm 0.7$ | $0.512 \pm 0.15$ |
| CST-Net | $\mathbf{2.071 \pm 0.5}$ | $\mathbf{0.254 \pm 0.08}$ | $\mathbf{2.016 \pm 0.6}$ | $\mathbf{0.296 \pm 0.09}$ | $\mathbf{1.907 \pm 0.5}$ | $\mathbf{0.297 \pm 0.09}$ | $2.700 \pm 0.6$ | $0.228 \pm 0.06$ | $\mathbf{2.295 \pm 0.6}$ | $\mathbf{0.313 \pm 0.11}$ | $\mathbf{2.243 \pm 0.5}$ | $\mathbf{0.413 \pm 0.12}$ |

Node2Vec (*Grover & Leskovec, 2016*), $p$, $q$ are chosen from {0.25, 0.5, 1, 2, 4}, the length of walk is chosen from {10, 25, 50, 80, 100}, and the candidate embedding size from {32, 64, 128}. For mini-batch k-means++ (*Arthur & Vassilvitskii, 2007*), batch size is selected from {1,000, 5,000, 10,000, 50,000} and the number of clusters from {32, 64, 128, 256}.

## Prediction performance

As shown in Table 2, the overall prediction performance of all competing methods across two datasets is displayed. Except for the prediction based on 3 year observation on APS dataset, which our model performs on par with the state-of-the-art method, the proposed CST-Net outperforms all four baseline methods on both scenarios. Because of the richer information within a longer observation time window, we can find that the longer the observation time, the better the prediction performance.

We noticed that the model of Feature-linear and Feature-deep perform not well in popularity prediction since it is difficult to extract all of the effective hand-crafted features of a cascade. Within the short observation time, the experimental result of DeepCas is better than Feature-linear and Feature-deep. Nevertheless, when the observation time being longer, Feature-linear and Feature-deep outperform DeepCas significantly. One possible explanation is that DeepCas focuses on sampling sequences *via* random walks while ignores the rich structural and temporal information of a cascade.

On Sina Weibo dataset, our model outperforms all of the four baseline methods within any observation window. However, we can see that CST-Net is slightly worse than DeepHawkes when the observation time window is set to 3 years on APS dataset. One possible explanation is that the community interaction matrix would be sparse because the life cycle of articles is significantly longer than that of microblogs. Therefore, it is difficult for CST-Net to capture the dynamics of citations. We leave exploring the sparsity problem to our future work.

It is worth mentioning that we have also investigated some poor prediction cases. Surprisingly, we found that there is an interesting number of cascades which do not get much retweets/citations for a period of time after being published, but then suddenly start getting popular heavily. This phenomenon is called "sleeping beauties" which has raise

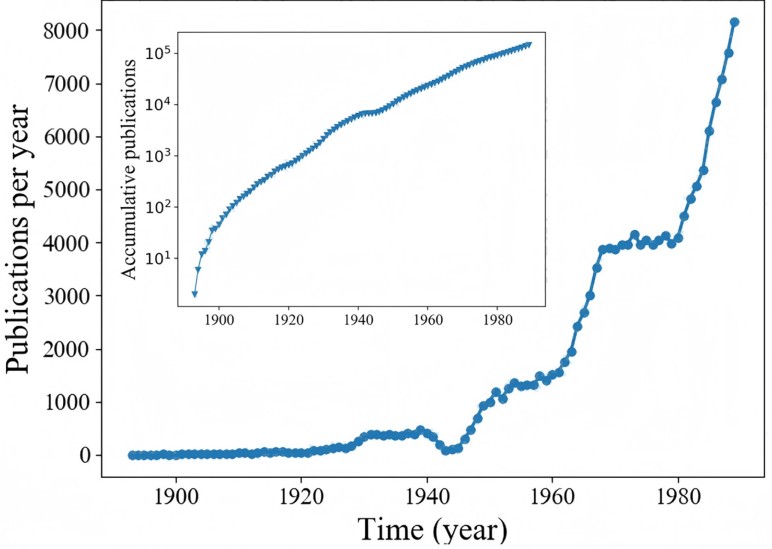

**Figure 6 The number of articles published each in APS.** Inset: accumulative number of articles published up to year $t$.     

much attention in other fields such as social computational science. However, how to effectively model and predict this kind of cascades is another line of research, which is beyond the focus of this manuscript and we will try to solve them in our future work.

In addition, comparing the experimental results in two scenarios, we can find that the CST-Net exhibits a better performance in Sina Weibo than in APS. We show in Fig. 6 the number of articles published in each year in the APS dataset. The inset of Fig. 6 gives a cumulative view, *i.e.*, total number of articles published up to a certain year, on a log-linear scale. Figure 6 is in good agreement with previous findings (*Wang, Song & Barabási, 2012*) that the number of articles published each year increases exponentially, indicating that the cascading pattern of articles is quite different from that of microblogs. Therefore, the model trained in the early stage would be inappropriate for the prediction in 20 years, leading to the slightly poor performance in APS dataset.

## Ablation study

In order to demonstrated the effectiveness of the components in the proposed CST-Net, a detailed analysis is given in this section. For comparison, three simplified variants of CST-Net, denoted as CST-basic, CST-feature and CST-community are represented, where one or two components are removed from the complete CST-Net model.

**CST-basic** model randomly arrange communities in the interaction matrix, in which the similarity between adjacent communities is uncertain and neglected. The hand-crafted temporal and structural features are also discarded. This version is constructed to offer a basic model for comparison.

**CST-feature** model includes the hand-crafted features which implicate the structural and temporal characteristics of a cascade on the basis of CST-basic. This

**Table 3 Prediction performance of variants of CST-Net.**

| | Sina Weibo | | | APS | | |
|---|---|---|---|---|---|---|
| T<br>Evaluation metric | 1 h<br>MSLE | 2 h<br>MSLE | 3 h<br>MSLE | 3 y<br>MSLE | 6 y<br>MSLE | 9 y<br>MSLE |
| CST-basic | 2.842 | 2.394 | 2.060 | 3.318 | 2.512 | 2.369 |
| CST-feature | 2.827 | 2.198 | 2.033 | 3.241 | 2.485 | 2.333 |
| CST-community | 2.399 | 2.083 | 1.969 | 2.964 | 2.361 | 2.263 |
| CST-Net | 2.071 | 2.016 | 1.907 | 2.700 | 2.295 | 2.243 |
| CST-Net w/o Conv | 2.216 | 2.064 | 1.955 | 2.811 | 2.342 | 2.258 |
| CST-Net w/o SE-ResUnit | 2.107 | 2.032 | 1.931 | 2.793 | 2.313 | 2.249 |

version is used to evaluate whether extracted features are effective for popularity prediction.

**CST-community** model uses Algorithm 1 to construct community interaction matrix on the basis of CST-basic. We use this version to verify the effectiveness of community interaction matrix construction.

The prediction results of these simplified versions of CST-Net are shown in Table 3. Comparing to the complete version of CST-Net model, all these three variants lead to certain degradation of performance in different extent. Firstly, we can observe that CST-basic and CST-feature have similar prediction performance, and CST-feature is slightly better than CST-basic. Accordingly, these hand-crafted features play a positive role on popularity prediction. Secondly, the version of CST-community outperforms both CST-basic and CST-feature with a significant reduction of prediction error, especially within short observation time window. Since the detected communities in CST-basic and CST-feature are arranged randomly, the local similarity between adjacent communities is uncertain and neglected. However, in order to feed into convolutional neural networks, the local similarity in the input matrix is required. Thus, the community interaction matrix construction contributes to learn the deep features of cascade among communities. For simplicity, in this article, we utilize the historic interactions among users to compute the community similarity and then assemble them orderly in the community interaction matrix. In addtion, we also extended our ablation study to more precisely isolate the impact of each key component of CST-Net. Specifically, we conducted additional experiments where we remove the convolutional architecture (CST-Net w/o Conv) and residual units (CST-Net w/o SE-ResUnit) from CST-Net, respectively. The extended ablation study results are also summarized in Table 3. These results further demonstrate the effectiveness of each component to the model's performance. Besides, we can also observe that the convolutional architecture performs more importance than the residual units in the model.

Moreover, in order to investigate the ability of the proposed CST-Net to predict popularity dynamics, we set the training period, *i.e.*, T, as 2 h and 6 years for Sina Weibo and APS dataset respectively, and then predict the incremental popularity every 3 h or

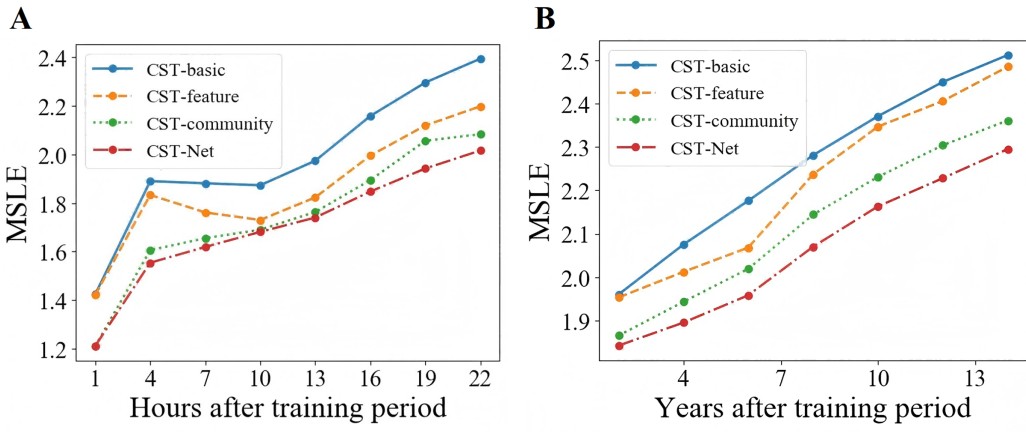

**Figure 7 Prediction performance of popularity dynamics.**

2 years on both scenarios. As shown in Fig. 7, we can observe that with the increase of prediction time, the performance of all four versions of CST-Net decreases.

## CONCLUSION

In this article, we propose an end-to-end deep learning framework for popularity prediction, called CST-Net, to combat the defects of existing methods. Firstly, we learn a low-dimensional embedding for users based on their historic interactions. Then, users are clustered into communities based on the learned user embeddings. Information cascades are represented as a series of episodes in the form of community interaction matrix. Afterwards, a convolutional architecture is applied to learn the representation of the entire cascade graph. Finally, the extracted structural and temporal features are further combined to predict the incremental popularity. We validate the effectiveness of CST-Net by applying it on two different types of population-scale datasets, *i.e.*, Sina Weibo and APS. Experimental results demonstrate that the proposed CST-Net model consistently outperforms the existing competitive popularity prediction methods. More importantly, it provides us great insights in understanding the fundamental mechanism of information diffusion and sheds light on the collective attention on online social networks.

There are a few limitations for the proposed method. Although the overall performance is pretty well, it does not hold for some abnormal dynamic processes with specific patterns or malicious behaviors. In addition, more flexible way of community interaction matrix construction can be investigated. Both of these are very interesting and we will try to solve them in our future work.

A long list of future work can be conducted for the follow-up researchers in this field. Examples include thorough investigation of the various roles played by individuals, deep exploration on the interplay between the dynamics of collective attention, and the structural and temporal characteristics of the networks spanned by early adopters. Specifically, how to effectively model the influence of users could provide us great insights in understanding the fundamental mechanism of information diffusion and shed light on

the collective attention on online social networks. It is also an interesting research topic to design an effective and efficient graph neural network to dynamically learn the concrete representation of the evolving cascade graph from different perspectives. In addition, one is also encouraged to investigate more implicit factors and devote to improve the robusarial of our proposed framework through generative adversarial networks or contrastive learning. Moreover, researcher could also dive deeper into the time-decaying effect modeling, as well as the model interpretability.

### Funding
This work was supported by the National Natural Science Foundation of China (No. 61263033, U21B2046). The funders had no role in study design, data collection and analysis, decision to publish, or preparation of the manuscript.

### Grant Disclosures
The following grant information was disclosed by the authors:
National Natural Science Foundation of China: 61263033, U21B2046.

### Competing Interests
The authors declare that they have no competing interests.

### Author Contributions
- Xuxu Zheng conceived and designed the experiments, performed the experiments, analyzed the data, performed the computation work, prepared figures and/or tables, authored or reviewed drafts of the article, and approved the final draft.
- Peng Bao conceived and designed the experiments, analyzed the data, authored or reviewed drafts of the article, and approved the final draft.
- Lin Qi performed the experiments, performed the computation work, prepared figures and/or tables, and approved the final draft.
- Chen Tian performed the experiments, performed the computation work, prepared figures and/or tables, and approved the final draft.
- Huawei Shen conceived and designed the experiments, authored or reviewed drafts of the article, and approved the final draft.

### Data Availability
The data and code are available at GitHub and Zenodo:
- https://github.com/BJTU-BAO/CST-NET.
- Peng, B. (2025). Cascade Dataset and Code [Data set]. Zenodo. https://doi.org/10.5281/zenodo.15428610.

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
