# Peer review of "CST-Net: community-guided structural-temporal convolutional networks for popularity prediction"

_PeerJ Computer Science, doi:10.7717/peerj-cs.2931_

## Round 0.1 · original submission · Major Revisions

Dear authors,

Thank you for submitting your manuscript. Feedback from the reviewers is now available. It is not recommended that your article be published in its current format. However, we strongly recommend that you address the issues raised by the reviewers, especially those related to readability, experimental design and validity, and resubmit your paper after making the necessary changes. Before submitting the manuscript, following should also be addressed:

1- The explanations about the organization of the paper are missing in the Introduction section.
2- Referencing style should be corrected.
3- To increase professionalism, references should be reinforced with recent papers from reputable journals. Recent literature should be deeply analyzed.
4- Space characters should be correctly used. See for example line: 106.
5- The values for the parameters of the algorithms should be provided.
6- The paper lacks the running environment, including software and hardware. The analysis and configurations of experiments should be presented in detail for reproducibility.
7- Equations should be used with correct equation number. Please do not use “as follows”, “given as”, etc. Explanation of the equations should also be checked. All variables should be written in italic as in the equations. Their definitions and boundaries should be defined. Necessary references should be provided.

Best wishes,

Reviewer 1 ·

Basic reporting

This paper presents an interesting approach for popularity prediction using community-guided structural-temporal convolutional networks (CST-Net). The method shows promising results on two large-scale datasets. However, there are a few areas that could be improved to strengthen the paper:

The motivation and novelty of the proposed CST-Net approach compared to existing deep learning methods for popularity prediction could be explained more clearly in the introduction. What are the key innovations?
More details should be provided on the community detection method used. How sensitive are the results to the choice of community detection algorithm?
The ablation study is helpful, but could be expanded to isolate the impact of each key component of CST-Net.
The discussion of limitations in the conclusion is good, but could be expanded with more concrete suggestions for future work to address these limitations.

Experimental design

The authors have conducted a comprehensive evaluation using two large-scale datasets from different domains (Sina Weibo and APS), which strengthens the generalizability of their findings. The comparison against several state-of-the-art baselines is appropriate and helps to contextualize the performance of CST-Net. The ablation study is a valuable addition, providing insights into the contribution of different components of the model. However, the experimental design could be improved in a few ways. First, the sensitivity of the results to key hyperparameters (e.g., number of communities, embedding size) should be explored and discussed. Second, while the MSLE and MRLE metrics are suitable, the addition of a ranking-based metric (e.g., NDCG) could provide a more comprehensive evaluation, especially given the importance of ranking in many real-world applications of popularity prediction. Lastly, the paper would benefit from a more detailed analysis of cases where CST-Net performs particularly well or poorly compared to baselines, which could provide insights into the strengths and limitations of the approach.

Validity of the findings

Consider including confidence intervals or statistical significance tests for the performance comparisons in Table 2. Additionally, a more detailed error analysis examining specific cases where CST-Net outperforms or underperforms compared to baselines could provide valuable insights into the model's strengths and limitations, enhancing the overall validity and interpretability of your results.

Additional comments

no comment

Reviewer 2 ·

Basic reporting

All comments have been added in detail to the last section.

Experimental design

All comments have been added in detail to the last section.

Validity of the findings

All comments have been added in detail to the last section.

Additional comments

Review Report for PeerJ Computer Science
(CST-Net: Community-guided structural-temporal convolutional networks for popularity prediction)

1. Within the scope of the study, a deep learning based model called CST-Net, which can perform popularity prediction operations, has been proposed using two different population-scale datasets.

2. In the introduction section, user generated content items and their importance, some popularity prediction studies based on deep learning and machine learning, the importance of the CST-Net proposed in the study, and its main contributions to the literature in three main aspects have been clearly mentioned at a sufficient level.

3. In the Related works section, studies related to popularity prediction have been examined in terms of different model methods such as generative and deep learning. In this section, the studies in the literature examined from three different perspectives should be compared in detail in terms of their differences, advantages, and shortcomings, and the deficiencies in the literature that the proposed model eliminates should be emphasized more clearly.

4. When the architecture of the CST-Net proposed within the scope of the study, the community interaction matrix construction component and algorithm, the convolutional architecture and user embedding components are examined in detail in order to perform popularity prediction operations, it is observed that it has the potential to contribute significantly to the literature and contains a certain level of originality.

5. It is stated that American Physical Society and Sina Weibo datasets are preferred as datasets. Explain in more detail the reasons why datasets containing cascades in paper citation and social networks are preferred, why these two datasets are preferred especially despite the fact that there are many different types of datasets in the literature, and the differences of other datasets.

6. The Mean Relative Log-transformed Error and Mean Square Log-transformed Error evaluation metrics obtained regarding the solution of popularity prediction problems are sufficient in terms of analysis of the results and type. Using these results from more than one different dataset using the proposed model highlights the usability of the model.

7. All metric values such as learning rate and batch size used in parameter settings should be explained in detail. Since the selection of these parameters and even the slightest change can positively or negatively affect the result, the details of the choices here are very important.

8. Although the results obtained with CST-Net seem sufficient, in order to compare the results with other deep learning based models, the determination of these other models should be detailed further. In addition, in order to highlight the superiority of the model, it is definitely recommended to compare the results with several other state-of-the-art models.

In conclusion, although the study proposes a deep learning based model with a certain quality of originality in terms of contributing to the literature in solving popularity prediction problems, all the parts mentioned above should be explained step by step by paying close attention and the necessary changes/updates should definitely be made in the paper.

---

## Round 0.2 · accepted · Accept

Dear Authors,

One of the previous reviewers did not respond to the invitation to review the revised manuscript. The other reviewer accepts your paper. I also think that your paper seems improved and ready for publication.

Best wishes,

Reviewer 2 ·

Basic reporting

All comments have been added in detail to the last section.

Experimental design

All comments have been added in detail to the last section.

Validity of the findings

All comments have been added in detail to the last section.

Additional comments

Review Report for PeerJ Computer Science
(CST-Net: Community-guided structural-temporal convolutional networks for popularity prediction)

Thanks for the revision. The new version of the paper and the answers given are generally at an appropriate level. I do not request any additional changes. It is sufficient for me as it is. I wish the authors success in their future projects. Best regards.